# Clinical Features of Transient Growth Hormone Deficiency

Yuki Sakai [1,*], Kento Ikegawa [2], Kazuhiro Shimura [3] and Yukihiro Hasegawa [2,*]

1   Department of Pediatrics, National Defense Medical College, Saitama 359-8513, Japan
2   Division of Endocrinology and Metabolism, Tokyo Metropolitan Children's Medical Center, Tokyo 183-8561, Japan
3   Department of Pediatrics, Keio University School of Medicine, Tokyo 160-8582, Japan
*   Correspondence: yukisakai36b@gmail.com (Y.S.); yhaset@gmail.com (Y.H.)

**Abstract:** Background: Most patients with idiopathic growth hormone deficiency (iGHD) in childhood have normal GH stimulation test results in adulthood. The present study aimed to investigate the characteristics and possible etiology of transient iGHD. Methods: Patients with childhood-onset iGHD who completed their GH treatment between March 2010 and March 2021 were retrospectively studied. Patients with a clear history of child abuse or constitutional delay of growth and puberty were excluded. Ten patients with a diagnosis of iGHD based on a decreased growth rate and growth hormone stimulation test findings at the time of onset were included. Retesting demonstrated that these patients had a normal GH level. Results: Five patients had insufficient weight gain (BMI SD score < −1.0 at the start of treatment or a decrease in BMI SD score > 1.0 from one year before treatment to the start of treatment). The other five patients had no remarkable clinical features. One patient had decreased height velocity at the same time as their sibling. Conclusion: Insufficient pre-treatment weight gain or a familial cluster of cases may be related to low GH peaks of GHST, leading to a diagnosis of transient GHD.

**Keywords:** pituitary; transient idiopathic growth hormone deficiency; growth hormone stimulation test

## 1. Introduction

Growth hormone deficiency (GHD) is defined as a disorder of inadequate growth hormone secretion from the pituitary gland. In children, the most common symptom is growth failure, and biochemical data indicating low GH secretion, such as inadequate responses on growth hormone stimulation tests (GHSTs) and low IGF-I values [1], are needed for diagnosis. A positive response to growth hormone replacement therapy is also a hallmark of this disorder. Childhood GHD is commonly idiopathic [2]. Most patients with idiopathic GHD (iGHD) during childhood reportedly have normal GH stimulation test results as adults [3,4]. These cases are referred to as transient iGHD in this study.

The etiology of transient iGHD is not clear, and the results of GHSTs for childhood iGHD are sometimes false positive owing to the influence of various factors, such as obesity [5–7], undernutrition [8], poor environment including child abuse [9,10], and delayed puberty [11–13]. However, iGHD may be transient in other cases without these factors. The present study aimed to investigate the characteristics and possible etiology of transient iGHD.

## 2. Materials and Methods

Patients with childhood-onset iGHD who completed their GH treatment between March 2010 and March 2021 at Tokyo Metropolitan Children's Medical Center were retrospectively studied. The diagnosis of GHD usually requires inadequate responses to two GHSTs. Given the limitations such as false positives and lack of reproducibility [1,2,14,15] of the GHST in diagnosing this condition, the definition of GHD in the present study required the fulfillment of all the following criteria:

1. pre-treatment height < −2.0 SD
2. pre-treatment height velocity < 0 SD
3. pre-treatment IGF-I value < −2.0 SD
4. at least two GHSTs with a peak value < 6.0 ng/mL
5. improvement in height velocity (>1.0 SD) after GH treatment

Transient GHD was defined by GH peak > 3.0 ng/mL or IGF-I > 150 ng/dL, the lower limits for young adults [16], on a retest, after GH interruption > one month. Patients with chromosomal abnormalities, brain lesions such as tumors, lymphocytic hypophysitis, and invisible pituitary stalk and ectopic posterior pituitary, a clear history of child abuse, or constitutional delay of growth and puberty (CDGP), were excluded. The average age at puberty onset in Japanese females and males is 10 and 11 years [17], respectively. Puberty onset is considered to be delayed if it occurs after age 12 years in females or after age 13 years in males. Poor weight gain was inferred from a BMI SD score < −1.0 at the start of GH treatment or a decrease in the BMI SD score > 1.0 from one year before, to the beginning of, treatment [18].

The Wilcoxon, Mann–Whitney U, and chi-square tests were used as appropriate with Easy *R* version 3.5.2 [19]. $p < 0.05$ was considered to indicate statistical significance. The present study complied with the 1964 Helsinki Declaration and its later amendments (in 2013) or comparable ethical standards. The Institutional Ethics Committee of Tokyo Metropolitan Children's Medical Center approved this study (No. 2021b-49).

## 3. Results

Of 98 patients receiving GHD at our hospital during the study period, 31 completed the treatment. Ten of the 31 patients had pre-treatment IGF-I > −2.0 SD. Eight patients with chromosomal abnormalities, brain lesions, a clear history of child abuse or CDGP, and two patients without follow-up were excluded. One patient had a low IGF-I level less than 150 ng/mL, −3.7 SD, at the time of GH discontinuation, but a retest was not yet performed. Ten patients with transient iGHD were finally enrolled (Figure 1). All patients underwent a hypothalamic–pituitary MRI to reveal no brain lesions. Three patients had permanent iGHD.

Table 1 shows the patients' clinical backgrounds. The median age at the start of treatment was 6.1 years, and the median height velocity was −4.1 SD. Insulin, arginine, clonidine, L-dopa, and glucagon were used for GHSTs. Five patients underwent another GHST after completing their treatment. The remaining five patients were judged not to have adult GHD based on adequate IGF-I levels.

**Table 1.** Clinical backgrounds.

| Patients with Transient iGHD (n = 10) | |
|---|---|
| Pre-treatment | |
| Age at the start of treatment | 6.1 (5.3~8.5) |
| Height velocity SD score | −4.1 (−4.9~−3.4) |
| Height SD score | −3.0 (−3.9~−2.8) |
| BMI SD score | −1.0 (−2.6~−0.3) |
| IGF-I SD score | −3.1 (−3.3~−2.3) |
| One year after the start of treatment | |
| Height velocity (cm/year) | 9.2 (8.0~10.2) |
| ⊿ SD Height | 0.7 (0.6~1.1) |
| ⊿ SD BMI | 0.1 (−0.1~0.4) |
| Post-treatment | |
| Age at the completion of treatment | 16.8 (14.7~17.4) |
| Treatment period | 10.6 (7.3~12.3) |
| IGF-I SD score | −1.3 (−1.6~−0.8) |

⊿ SD Height: Improvement in height SD score in the first year after treatment start; ⊿ SD BMI: Improvement in BMI SD score in the first year after treatment start. Results are expressed as median (first quartile–third quartile).

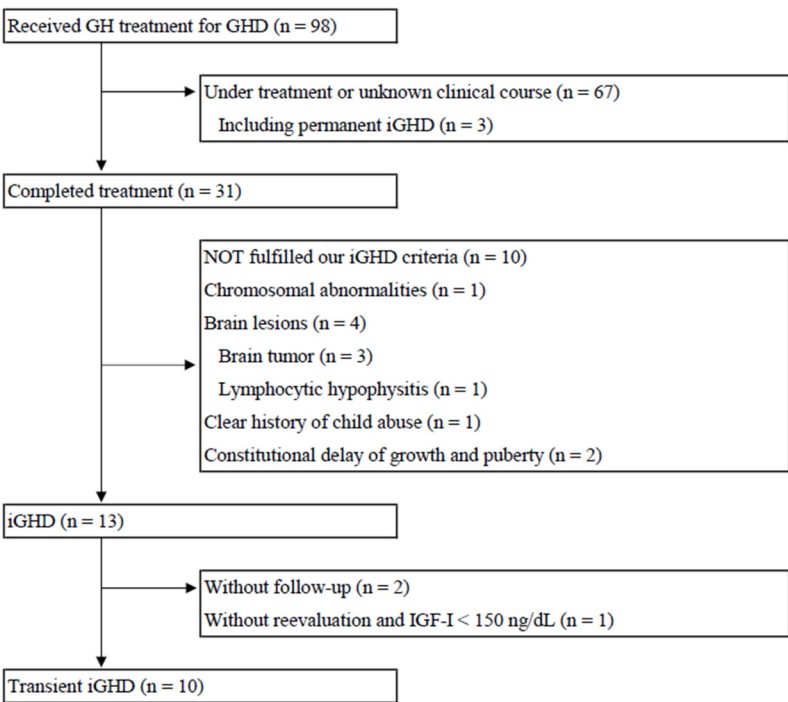

**Figure 1.** Patient selection. iGHD criteria: (1) pre-treatment height < −2.0 SD, (2) pre-treatment height velocity < 0 SD, (3) pre-treatment IGF-I < −2.0 SD, (4) at least two GHSTs with a peak value < 6.0 ng/mL, and (5) improvement of height velocity (>1.0 SD) after GH treatment.

Five patients had insufficient weight gain before GH treatment and were placed in a low BMI group. The other five patients had no remarkable clinical features and were labeled as a non-low BMI group. One patient had decreased height velocity at the same time as their sibling. Figure 2 shows their growth curves. Another representative four growth curves are in the Supplemental Figure S1: two cases each for both low BMI and non-low BMI groups.

Table 2 compares the low-BMI group with the non-low-BMI group. There was no significant difference in height SD score or height velocity between the groups. The BMI SD tended to increase after treatment in the low-BMI group. The average peak value of GHSTs was not significantly different between the two groups. In addition, three patients receiving a hormone replacement therapy other than GH (one receiving levothyroxine [LT4] replacement therapy and two receiving LT4 and hydrocortisone [HDC] replacement therapy) were included in the low-BMI group. Two of these patients completed their LT4 and/or HDC therapy.

**Table 2.** Comparison between low-BMI group and non-low-BMI group.

|  | **Low BMI (n = 5)** | **Non-Low BMI (n = 5)** |
|---|---|---|
| Pre-treatment |  |  |
|   Age at the start of treatment | 8.9 (1.8~11.3) | 5.5 (5.3~6.7) |
|   Height velocity SD score | −4.8 (−5.2~−3.6) | −4.1 (−4.1~−3.4) |
|   Height SD score | −3.1 (−4.1~−2.8) | −2.9 (−3.4~−2.8) |
|   BMI SD score * | −2.8 (−2.9~−2.2) | 0.3 (−0.4~0.4) |
|   IGF-I SD score | −3.1(−3.1~−2.2) | −3.3 (−3.6~−2.3) |
|   Average peak value of GHSTs | 3.4 (2.7~4.3) | 3.6 (3.5~3.9) |
|   Two GHSTs with a peak value < 3 ng/mL | n = 1 | n = 1 |
| One year after the start of treatment |  |  |
|   Height velocity (cm/year) | 9.3 (7.9~10.3) | 9.1 (8.1~10.0) |
|   ⊿ SD Height | 0.6 (0.4~1.1) | 0.8 (0.6~1.0) |
|   ⊿ SD BMI | 0.5 (0.4~1.6) | 0.1 (−1.0~0.1) |
| Post-treatment |  |  |
|   IGF-I SD score | −1.2 (−1.7~−1.2) | −1.4 (−1.6~−0.7) |
|   Other hormone replacement | n = 3 | n = 0 |

⊿ SD Height: Improvement in height SD score in the first year after treatment start; ⊿ SD BMI: Improvement in BMI SD score in the first year after treatment start; results are expressed as median (first quartile ~ third quartile); *: $p < 0.05$.

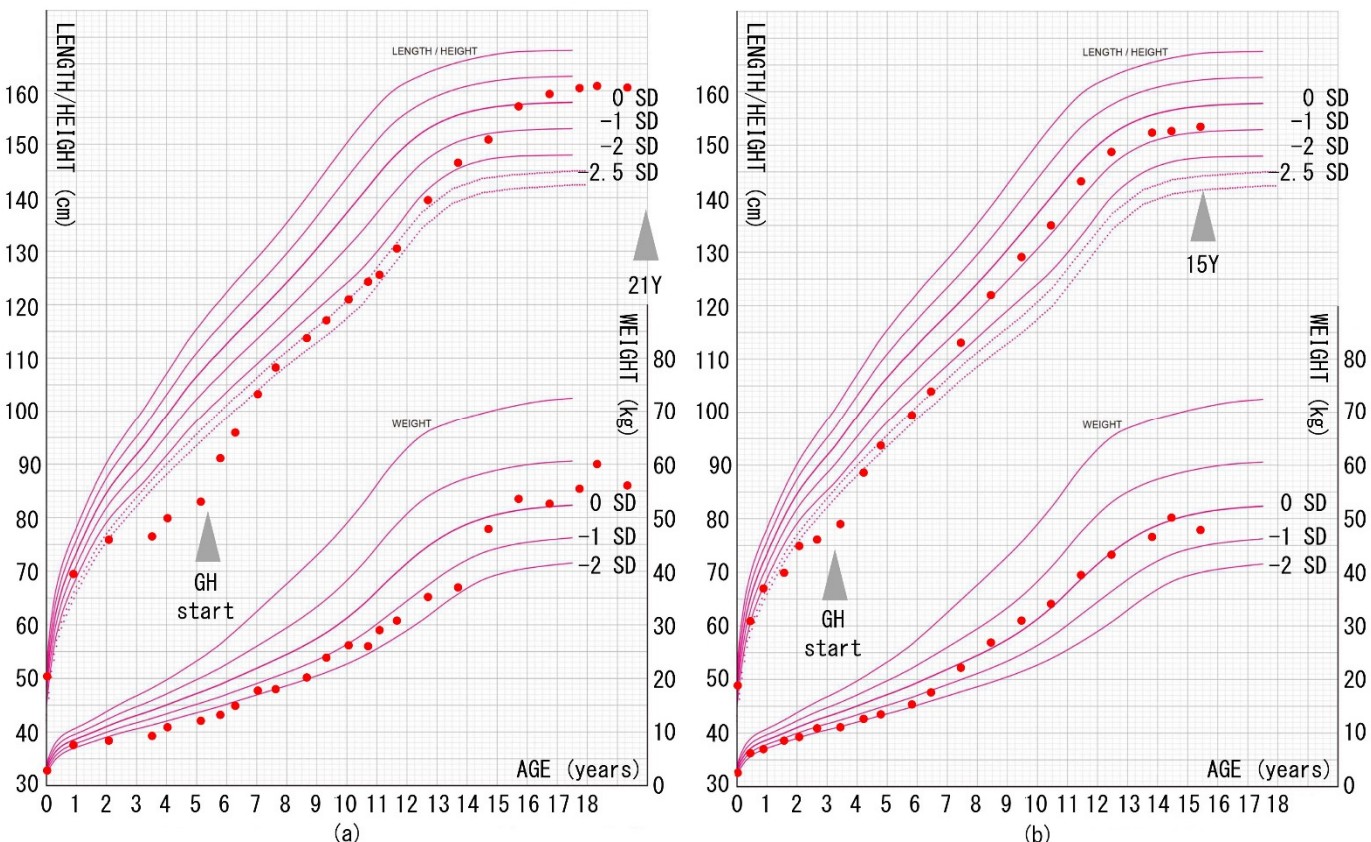

**Figure 2.** Growth chart of transient growth hormone deficiency in two siblings. Arrows show the start and the completion of GH replacement therapy. (**a**) Older sister. (**b**) Younger sister. The latter had severe short stature (−4.5 SD), a low IGF-I level (−3.6 SD), and decreased height velocity (−5.8 SD) at the start of treatment but only one GHST was positive. She was clinically diagnosed and treated (She was not included in this study). The charts were referenced from [20].

## 4. Discussion

The present, retrospective study was the first to analyze the etiology of transient iGHD and found that five of ten patients with transient iGHD had insufficient, pretreatment weight gain and post-treatment BMI SD tended to increase. The clinical feature of the iGHD in the remaining five patients was unknown, but one was part of a familial cluster, suggesting the possible involvement of environmental factors. However, this does not necessarily exclude the possibility of genetic causes; as far as we know, no genetic cause is proved to be associated with transient iGHD.

The reasons for the normalization of GHST findings in adulthood are unknown. On the other hand, CDGP and poor environment may account for an inadequate response on a GHST [9–13]. A previous study reported that 40–80% of patients with isolated GHD had normal GHST results when retested [21–26]. The high false-positive rate of the GHST in approximately 20–30% of patients [14] may also be related to the apparent normalization. The prevalence of transient iGHD was 76% (10/13) in this study, which was consistent with the previous findings. Table 3 shows the previously reported prevalence of transient GHD. The characteristics of the subject in this study differ from those in previous reports are the inclusion of a considerable number of low-BMI patients and the inclusion of idiopathic multiple pituitary hormone disorder. In the present study, the initial GHD diagnosis was not based only on the GHST results but also relied on auxological data and IGF-I values.

**Table 3.** The prevalence of transient GHD.

| | tGHD (%) | Reevaluation | Subject | Distinctive Criteria * |
|---|---|---|---|---|
| M Tauber et al., 1997 [21] | 81/121 (67) | During the first year after completion of treatment | Idiopathic GHD Auxological criteria was not described | Excluding malnutrition |
| Loche S et al., 2000 [22] | 28/33 (84) | One to six months after diagnosis None of the children received treatment | Isolated and idiopathic GHD Height $\leq -2$ SD Decreased HV (<25th %tile) SS relative to mid-parental height | Exclusion criteria was not described |
| Zucchini S et al., 2006 [23] | 25/69 (36.2) | Four to six weeks after interruption of treatment Testicular volume 6–12 mL (males) Breast development 2–3 (females) | Isolated and idiopathic GHD Height < 3rd %tile Height < 10th %tile with TH > 50th %tile Decreased HV < 25th %tile No signs of puberty | Excluding MPHD and age at diagnosis < 6 year |
| Smyczyńska J et al., 2014 [24] | 132/150 (88) | At least one month after completion of treatment | Isolated and idiopathic GHD Height < 3rd %tile | Excluding MPHD |
| Vuralli et al., 2017 [25] | 69/170 (40.6) | At the end of the first year of treatment | Isolated GHD Height < −2 SD Decreased HV (<25th %tile) Delayed BA (2 SD below the CA) | Including 21 patients with brain tumor |
| L Penta et al., 2019 [26] | 26/31 (83.9) | At least three months after completion of treatment | Isolated and idiopathic GHD Auxological criteria was not described | Excluding MPHD |

*: This is not all inclusive. HV: height velocity; SS: short stature; TH: target height; MPHD: multiple pituitary hormone deficiency; BA: bone age; CA: chronological age.

Although the patients with a clear history of child abuse were excluded, about half the enrolled patients had low BMI values. It was impossible to determine whether the low BMI in our patients was caused by undernourishment or if other, environmental factors were involved. Undernourishment usually leads to low IGF-I levels owing to increased GH resistance and high basal GH levels owing to release from feedback inhibition [27]. However, all the patients in the low BMI group responded well to treatment, suggesting

their GH resistance was mild. In line with our results, a previous study demonstrated a low basal GH level in malnourished children [28]. Patients with emotional deprivation also have clinical features of GHD and sometimes have biochemical profiles resembling that of panhypopituitarism [9,10,29].

In recent years, several studies have discussed the timing of GHD reassessment [22,23,25]. Treatment should not be administered indiscriminately, and early treatment completion alleviates the burden on the patients. Opinions about when retesting should be performed vary. The indicators for early reevaluation are GHD severity, age at diagnosis, other hormone replacement therapies, and a rapid, post-treatment height increase [23,25]. Given that most patients with iGHD have the transient form of the disease and that it is difficult to identify the predisposing factors in the patients' background, re-evaluating all patients with iGHD may be a prudent measure. In the present study, about half of the iGHD patients had a low BMI at the start of treatment. This fact could be an indication for an early GHST following GH interruption.

LT4 therapy was discontinued in two of the three patients receiving LT4 with GH. These two patients were in the low-BMI group; thus, the need for LT4 therapy might also be connected to low BMI or some other, environmental factor. Several reports have shown that biochemical findings similar to those of central hypothyroidism may be associated with malnutrition, such as that caused by anorexia nervosa [30,31]. The effects of malnutrition on the adrenocorticotropic hormone (ACTH) level are not well known. Patients with anorexia nervosa have increased basal cortisol levels but a decreased response on ACTH stimulation tests. [32,33]. Patients with emotional deprivation, as described above, have biochemical data resembling those of panhypopituitarism, including ACTH deficiency [9].

The strength of the present study is its inclusion of a relatively large number of patients who had attained an adult height and were confirmed not to have adult GHD. The study also defined the diagnostic criteria for childhood iGHD strictly. One limitation of the present study was its inability to determine the cause of insufficient weight gain. Additionally, some of the patients were unable to be followed-up after treatment and were therefore not included.

## 5. Conclusions

About half the patients with transient iGHD had poor weight gain at the start of treatment. Some factors causing low BMI may influence GHST, so interpretation in patients with low BMI should be cautious in diagnosing GHD. Patients with iGHD should undergo reevaluation before reaching their final height to help avoid unnecessary treatment.

**Supplementary Materials:** The following supporting information can be downloaded at: https://www.mdpi.com/article/10.3390/endocrines4010009/s1, Figure S1: Growth chart of two cases each for both low BMI and non-low BMI groups.

**Author Contributions:** Conceptualization, Y.S., K.I., K.S. and Y.H.; data curation, Y.S. and K.S.; writing the first draft preparation, Y.S. and Y.H.; manuscript review and editing, Y.H. All authors have read and agreed to the published version of the manuscript.

**Funding:** This research received no external funding.

**Institutional Review Board Statement:** The study was conducted according to the guidelines of the Declaration of Helsinki, and approved by the Institutional Review Board (or Ethics Committee) of Tokyo Metropolitan Children's Medical Center (No. 2021b-49; 19 August 2021).

**Informed Consent Statement:** Not applicable.

**Data Availability Statement:** The data used in this study are available on reasonable request from the corresponding author.

**Acknowledgments:** We are indebted to James R. Valera for his assistance with editing this manuscript.

**Conflicts of Interest:** The authors declare no conflict of interest.

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
