# Peer review of "Clinical Features of Transient Growth Hormone Deficiency"

_endocrines, doi:10.3390/endocrines4010009_

Round 1

Reviewer 1 Report

The authors investigated the characteristics and possible etiology of transient isolated growth hormone deficiency.  I have the following comments:

1. Line 45: the pre-treatment height velocity < 0 SD seems to be weak criteria.

2. Line 46: IGF-1 <- 1.0 SD is also weak criteria

3. Line 47: If patient fails one of GHST but passes the other one, patient does not have growth hormone deficiency.

4. Line 54: criteria for delayed puberty are not standard.

5. Line 78: what agents were used for GHST

6: Line 122 and conclusions --conflict.   If patients have malnutrition, they will have growth hormone resistance not growth hormone deficiency.  Then why they failed GHST.  Low BMI does not always mean undernutrition.

Author Response

Thank you for reviewing our paper constructively. We have answered each of your points below.

Comment 1-3:

Line 45: the pre-treatment height velocity < 0 SD seems to be weak criteria.

Line 46: IGF-1 <- 1.0 SD is also weak criteria

Line 47: If patient fails one of GHST but passes the other one, patient does not have growth hormone deficiency.

RESPONSE:

We changed the biochemical criteria into the followings: pre-treatment IGF-I value < -2.0 SD, at least two GHSTs with a peak value < 6.0 ng/mL (Page 2, Line 49-50).

We think the auxological criteria of height velocity (HV) < 0 SD is not so weak. This value is not Ht SDS but HV SDS. If you have a different thought, just let me know.

Comment 4:

Line 54: criteria for delayed puberty are not standard.

RESPONSE:

As shown in Page 2, Line 55-7, the average puberty age in Japan is younger than in the U.S. and most of Europe. One SD is roughly equal to one year (Ref 17). When the onset of puberty is delayed by two years (2 SD) more than average, we consider it delayed.

Comment 5:

Line 78: what agents were used for GHST

RESPONSE:

We added the sentence as follows: Insulin, arginine, clonidine, L-dopa, and glucagon were used for GHSTs (Page 3, Line 79-80).

Comment 6:

Line 122 and conclusions --conflict.   If patients have malnutrition, they will have growth hormone resistance not growth hormone deficiency. Then why they failed GHST.  Low BMI does not always mean undernutrition.

RESPONSE:

The patients in this study were not in typical "malnutritional conditions."

We agree that malnutrition is not always the primary cause of low BMI. As far as we know, low BMI and growth hormone resistance have not been linked. To emphasize these things, we moved the explanation of low BMI from the 4th paragraph (Page 6, Line 158) to the 3rd paragraph (Page 5, Line 139-41) in the discussion.

Reviewer 2 Report

The commentary by Sakai et al. describes the features of clinical transient GH deficiency in patients. The manuscript are well-written and the discussion is sensible.

Below are some specific comments to improve the manuscript:

Abstract:

·      GHD should be introduced before being abbreviated in the abstract.

Material and Methods:

·      “Given the limitations [1,2] of the GHST in diagnosing this condition”. For clarity, please specify these limitations.

Results:

·      Figure 2 depicting growth charts is very useful but the small details and legends are impossible to see when they are so small? Consider arrange the two graphs on top of each other to be able to increase their size.

·      LT4 and HDC should be introduced before being abbreviated.

Discussion:

·      “The etiology of the iGHD in the remaining nine patients was unknown, but three were part of a familial cluster, suggesting the possible involvement of environmental factors.” Could genetic factors also play a role for these patients since they are clustered. Please discuss?

·      “…child abuse may account for an inadequate response on a GHST.” Please elaborate on the relationship between child abuse and non-sufficient GHST response?

·      ACTH should be introduced before being abbreviated.

Author Response

We appreciate your excellent and generous review of our paper. We have answered each of your points below.

Comment 1:

Abstract: GHD should be introduced before being abbreviated in the abstract.

RESPONSE:

We added the term and its abbreviation as suggested (Page 1, Line 9).

Comment 2:

Material and Methods: “Given the limitations [1,2] of the GHST in diagnosing this condition”. For clarity, please specify these limitations.

RESPONSE:

As suggested, we mentioned false positives and lack of reproducibility as the limitations of the GHST (Page 1, Line 44).

Comment 3:

Results: Figure 2 depicting growth charts is very useful but the small details and legends are impossible to see when they are so small? Consider arrange the two graphs on top of each other to be able to increase their size.

RESPONSE:

We changed the figures and text to make them more legible (Page 4).

Comment 4:

Result: LT4 and HDC should be introduced before being abbreviated.

RESPONSE:

We added the terms and their abbreviations as suggested (Page 3, Line 98-9).

Comment 5:

Discussion: “The etiology of the iGHD in the remaining nine patients was unknown, but three were part of a familial cluster, suggesting the possible involvement of environmental factors.” Could genetic factors also play a role for these patients since they are clustered. Please discuss?

RESPONSE:

Thank you for pointing out the critical issue. We added the sentence as follows: However, this does not necessarily exclude the possibility of genetic causes; as far as we know, no genetic cause is proved to be associated with transient iGHD (Page 4, Line 119-21).

Comment 6:

Discussion: “…child abuse may account for an inadequate response on a GHST.” Please elaborate on the relationship between child abuse and non-sufficient GHST response?

RESPONSE:

False positives of the GHST have been linked to poor environments, such as child abuse (Ref 9, 10). We edited the key sentences on this issue in the introduction (Page 1, Line 36) and the discussion (Page 4, Line 123).

Comment 7:

Discussion: ACTH should be introduced before being abbreviated.

RESPONSE:

We added the term and its abbreviation as suggested (Page 6, Line 164).
